



# Gravity Inversion Method to Produce Compact and Sharp Images Using $L_0$-norm Constraint with Auto-adaptive Regularization and Combined Stopping Criteria.

Mesay Geletu Gebre [1][*] and Elias Lewi [2]

[1]Wolkite University, College of Natural and Computational Science,

P.O.Box 07, Wolkite, ETHIOPIA, E-mail: mesaygeletu@gmail.com

[2]Addis Ababa University, Institute of Geophysics, Space Science and Astronomy, Addis Ababa, ETHIOPIA

---

[*]Mesay Geletu, Email: mesaygeletu@gmail.com



## Abstract

We present a gravity inversion method that can produce compact and sharp images, to assist the modeling of non-smooth geologic features. The proposed iterative inversion approach makes use of $L_0$-norm stabilizing functional, hard, and physical parameter inequality constraints, and depth weighting function. The method incorporates an auto-adaptive regularization technique, which automatically determines a suitable regularization parameter and error weighting function that helps to improve both the stability and convergence of the method. The auto-adaptive regularization and error weighting matrix are not dependent on the known noise level. Because of that, the method yields reasonable results even the noise level of the data is not known properly. The utilization of an effectively combined stopping rule to terminate the inversion process is another improvement that is introduced in this work. The capacity and the efficiency of the new inversion method were tested by inverting randomly chosen synthetic and measured data. The synthetic test models consist of multiple causative blocky bodies, with different geometries and density distributions that are vertically and horizontally distributed adjacent to each other. Inversion results of the synthetic data show that the developed method can recover models that adequately match the real geometry, location, and densities of the synthetic causative bodies. Furthermore, the testing of the improved approach using published real gravity data confirmed the potential and practicality of the method in producing compact and sharp inverse images of the subsurface.

*Keywords*— Gravity data, Iterative inversion, $L_0$-norm constraint, Auto-adaptive regularization, Stopping criteria, Compact image.



## 1 Introduction

Gravity measurements have been used in a wide range of geophysical prospecting and investigations, such as in mineral explorations, engineering and environmental problems as well as archeological site investigations (Hinze et al., 2013, p. 20). In general, gravity inversion is a process that is used to determine the density, size, shape, and location of a complex subsurface causative bodies from an observed gravity anomaly, by using different mathematical modeling techniques. Thus, inversion of gravity data constitutes an important step in the quantitative interpretation since the reconstruction of density contrast models markedly increases the amount of information that can be extracted from the gravity data.

However, a principal difficulty with the gravity data inversion is the inherent non-uniqueness and instability that also exists in any geophysical method (Al-Chalabi, 1971; Blakely, 1996, p. 216). In other words, for the given observed gravity data there are many equivalent density distributions that can reproduce the same field data. The standard approach used to select acceptable solutions, that are geologically reasonable, is to use an additional information about the problem by making assumptions on the following aspects: (1) about the model parameters (existing information on the subsurface structure from geological or other geophysical hindsight) and (2) about the data parameters (statistical properties of the inexact data, e.g. Gaussian distribution of errors). Based on these assumptions there are two approaches in gravity inversion: The first approach fixes the density and vary the geometry. This approach is nonlinear in nature and has been studied by many authors, for instance, Lelievre et al. (2015); Camacho et al. (2002) and Camacho et al. (2011). The second approach, which also is the one used in this work, fixes the geometry and vary the density. This approach is linear in nature and has been investigated by many researchers (Li & Oldenburg, 1998; Boulanger & Chouteau, 2001).

In an effort to introduce more qualitative prior information, Last & Kubik (1983) in particular, developed a method called compact gravity inversion. Their strategy utilizes the compactness stabilizer to minimize the area (in 2D) or volume (in 3D) occupied by the causative body, which is equivalent to maximizing its compactness. Barbosa & Silva (1994) generalized the compact inversion method by making use of compactness along several axes using Tikhonov's regularization. In 2006 Silva and Barbosa further developed the Compact inversion method with so-called 'interactive inversion' which estimates the location and geometry of several density anomalies. They simplified their old method (Barbosa & Silva, 1994) to improve computational performance. The generalized compact and interactive inversion strongly need a priori information to yield an accurate estimation.

The compactness stabilizer (Last & Kubik, 1983) also known as the minimum support stabilizer (Port-



niaguine & Zhdanov, 1999) has been borrowed and implemented by other researchers in various geo-
physical inversion methods (Ajo-Franklin et al., 2007; Stocco et al., 2009; Fei et al., 2018; Feng et al.,
2020; Varfinezhad et al., 2020). As it was demonstrated by a number of researchers (Zhdanov & Tol-
staya, 2004; Rezaie et al., 2017; Feng et al., 2020; Varfinezhad et al., 2022), this stabilizer is known to
yield a compact or focused geophysical model with sharp boundaries. Apart from the inversion methods
which produce focused images mentioned above, sparse geophysical inversion approaches derived from
$L_p$-norm ($0 \leq p \leq 1$) stabilization have been developed by many researchers. For instance, sparse
seismic reflectivity inversion method (Li et al., 2017), direct current resistivity data inversion algorithm
(Singh et al., 2018), magnetic data sparse inversion method (Li et al., 2018; Fournier et al., 2020), sparse
gravity data inversion technique (Vatankhah et al., 2017; Peng & Liu, 2021), to mention only a few.
Some instability of the original compact gravity inversion algorithm of Last & Kubik (1983) was re-
ported by Lewi (1997, p. 87) when the data is contaminated with noise. Then Lewi (1997, p. 89)
has improved the original compact inversion by introducing a new approach to the 3D compact gravity
inversion. The problem with Lewi (1997, p. 89) method arises when dealing with a multiple-source
model, where the inversion algorithm tends to concentrate densities towards the surface regardless of
the true depth of the causative bodies. In overcoming this drawback, Gebre & Lewi (2022) improved
the compact gravity inversion method by incorporating a new depth weighting function. In this paper,
we present a gravity inversion method that can produce compact and sharp images, to assist the mod-
eling of non-smooth, blocky geologic features with sharp boundaries. The proposed approach is based
on the authors previous work (Gebre & Lewi, 2022), to which the reader is referred for further details,
with the following two main differences and advancements. The first is proposing and incorporating an
auto-adaptive regularization and error weighting function. This has improved the fast convergence of
the method while keeping its stability. The second is the implementation of combined stopping criteria
to terminate the iteration after an appropriate number of steps.

## 2 Methodology

### 2.1 The 2D model

Most fixed geometry gravity inversion algorithms, including the one presented here, employ rectangu-
lar prismatic elements, to discretize the subsurface, owing to their flexibility in constructing complex
models (Silva & Barbosa, 2006; Commer, 2011; Grandis & Dahrin, 2014). A 2-D model is obtained





by discretization of the subsurface under the survey area into a large number of infinitely long hori-
zontal rectangular prisms, with the infinitely long dimension oriented in the invariant y-direction, with
variations in densities only assumed for the X and Z directions. The 2-D model is illustrated in Fig.
1. The density contrasts are constant inside each cell only and can vary individually. Here we have
used equal dimension for the cells. However, the algorithm is flexible, to accommodate non-regular
size cells. Gravity stations indicted by ▽ symbols are located at the centers of the upper faces of the
rectangular blocks in the top layer. This discretization scheme of the subsurface allows us to calculate

the gravitational attraction caused by each rectangular block separately.

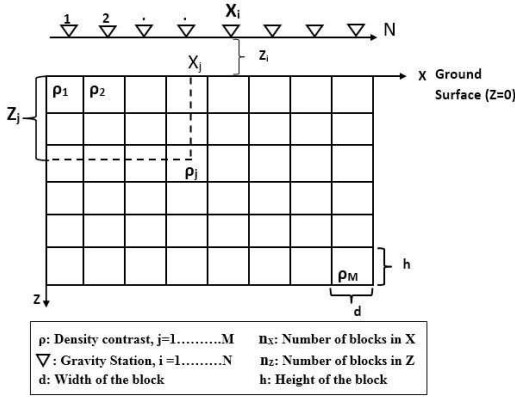

Figure 1: A 2-D model of the subsurface under a gravity profile. Gravity stations ($X_i$) are located at the centers of the blocks, indicated by the ▽ symbols.


## 2.2 Forward modelling

After discretization of the modeling space into a set of elementary rectangular blocks, the total vertical
component gravity response calculated at the $i^{th}$ observation point $g_i$ is the sum of the gravity contribu-
tions generated by each of the individual rectangular element, on all points belonging to the observation
grid and it is given by:

$$g_i = \sum_{j=1}^{M} a_{ij}\rho_j + e_i \quad i = 1, 2, 3....N \tag{1}$$

where $\rho_j$ is the density of the $j^{th}$ prism; $N$ denotes the numbers of observations; $a_{ij}$ is the contribution
of $j^{th}$ prism to the gravity value on $i^{th}$ observation point and $e_i$ is the noise associated with $i^{th}$ data point.
The kernel $a_{ij}$ is the forward operator that maps from the physical parameter space to the data space.
The exact mathematical expression of the kernel used here is presented by Last & Kubik (1983) which



is adopted from Nagy (1966) to which the reader is referred for more detail mathematical development.
In matrix notation Eq. (1) can be written as:

$$g = A\rho + e \tag{2}$$

where $g$ is an N-dimensional vector containing the gravity values, $\rho$ is an M-dimensional model vector
of densities, $A$ is the N x M kernel matrix, and $e$ represents the noise vector at data points. Equation
(2) constitutes the gravity forward modeling, i.e. used to calculate the predicted gravity anomalies
(theoretical data) for a known subsurface density contrast (model $\rho$).

## 2.3  Inverse Modeling

Our objective in solving gravity inverse problems is given the observed gravity data ($g$), we seek a
solution that gives a density distribution $\rho$ which predicts the observed data with a certain noise level
and at the same time, satisfies certain constraints. For the model presented here, the density vector
$\rho$ is related to the predicted gravimetric field $g$ by the linear expression given in Eq. (2). Like the
majority of practical inverse problems arising in geophysical modeling gravity inversion is an ill-posed
problem. Moreover, usually we have less number of the observed gravity data than the number of the
model parameters which makes the system under-determined problem. A standard way to solve such
ill-posed and under-determined problems, according to regularization theory (Tikhonov et al., 2013), is
minimization of the following objective function ($\Phi$) which is the combination of data fidelity or misfit
functional ($\Phi_d$) and stabilizing functional (stabilizer) term ($S(\rho)$):

$$\Phi = \Phi_d + \ell^2 S(\rho) \tag{3}$$

Here the misfit functional $\Phi_d = \|W_e(A\rho - g^{obs})\|_2^2$, and $W_e$ is error weighting diagonal matrix. In Eq.
(3), $\ell$ is a regularization parameter that controls the trade-off between the data fidelity and the stabilizing
term. Choosing a small value improves the data fit but the recovered models have highly oscillatory
artificial structures (which is equivalent to under-regularization). On the other hand, a large value of $\ell$
leads to a large misfit value between the observed and predicted data and a small norm of the model
(over-regularizing the solution). Thus, the choice of a suitable value for $\ell$ is very important.
The choice of the stabilizing functional, in Eq. (3), depends on the desired model features that are to
be recovered. There are several types of stabilizers that have been developed and implemented in the





inversion of potential field data, which can roughly be divided into two categories: (I) Smooth stabilizer
which uses $L_2$ -norm of the model parameters or gradient of the model parameters (Li & Oldenburg,
1998; Cella & Fedi, 2012; Paoletti et al., 2013). (II) Non- smooth stabilizer which uses $L_1$-norm or
$L_0$-norm directly on the model parameters or on the gradient of the model parameters (Bertete-Aguirre
et al., 2002; Sun & Li, 2014; Li et al., 2018; Utsugi, 2019). Inversion methods that utilize a smooth stabi-
lizer produce model typically characterized by smooth features, and hence have difficulties in recovering
blocky structures or non-smooth distributions that have sharp boundaries or abrupt changes in physical
properties (Farquharson, 2008). To overcome this problem, non-smooth stabilizers that help to produce
compact and sharp models have been applied successfully (Zhdanov, 2009; Meng et al., 2018). Since
we are interested in developing a gravity inversion method that can produce compact and sharp models,
we use a non-smooth stabilizer through the $L_0$-norm on the model parameters and will be discussed in
the next subsection.
Minimizing the objective function $\Phi$ in Eq. (3), using the standard weighted-damped least-square opti-
mization, the estimated density distribution in matrix notation can be given by (Menke, 1989, p. 55):

$$\boldsymbol{\rho}^{k+1} = \boldsymbol{\rho}_F^k + \left[ [\boldsymbol{W}_c^k]^{-1} \boldsymbol{A}^T \left( \boldsymbol{A}[\boldsymbol{W}_c^k]^{-1} \boldsymbol{A}^T + \ell^2 [\boldsymbol{W}_e^k]^{-1} \right)^{-1} \boldsymbol{g}_r^k \right] \qquad (4)$$

where the superscript $k$ denotes that variable at $k^{th}$ iteration and $\boldsymbol{W}_c^k$ is a combined weighting matrix.
$\boldsymbol{\rho}_F^k$ is reference density vector, which is from prior information or calculated at each iteration. $\boldsymbol{g}_r^k =$
$\boldsymbol{g}^{obs} - \boldsymbol{A}\boldsymbol{\rho}_F^k$ represents residual data vector computed at each iteration. Computation of the regularization
parameter $\ell$ in Eq. (4) will be described in Sect. 2.3.3. In this work, the combined weighting matrix
$(\boldsymbol{W}_c^k)$ is defined as a product of three different diagonal matrices, $L_0$-norm constraint matrix $(\boldsymbol{W}_{L_0}^k)$,
depth weighting $(\boldsymbol{W}_z)$ and hard constraint matrix $(\boldsymbol{W}_h^k)$.

$$\boldsymbol{W}_c^k = \boldsymbol{W}_{L_0}^k \boldsymbol{W}_z \boldsymbol{W}_h^k \qquad (5)$$

### 2.3.1   $L_0$-norm Constraint

The $L_0$-norm is commonly defined as the number of nonzero elements in a vector. Because there is no
analytical formula that meets the mathematical requirement to be regarded as $L_0$-norm, the approximate
expression is usually used to convert the $L_0$-norm into an equivalent norm for the suitability of computa-
tion. In literature (Zhao et al., 2016; Li & Yao, 2020) that discusses the inversion of potential field data,





different $L_0$-norm approximate stabilization functions have been developed and implemented to obtain
focused images and sharp boundaries. Meng (2016) used a hyperbolic tangent function to approximate
the $L_0$-norm and applied it for 3D inversion of gravity gradient tensor data. Meng et al. (2018) proposed
an exponential mathematical function to approximate the $L_0$-norm for 3D gravity sparse inversion. In
this paper, the minimum support functional, which is also called compactness constraint originally pro-
posed by Last & Kubik (1983) and then further extended by Portniaguine & Zhdanov (1999) to include
a reference model is selected which can be expressed as follows:

$$S(\rho) = \sum_{j=1}^{M} \frac{(\rho_j - \rho_j^{apr})^2}{(\rho_j - \rho_j^{apr})^2 + \varepsilon} \tag{6}$$

In our case to avoid the requirement of a prior model, we set $\boldsymbol{\rho}_j^{apr} = 0$. Using the function in Eq. (6)
as stabilizing functional in the objective function ($\Phi$) is equivalent to using $L_0$-norm based stabilization
and thus it can be rewritten as follows (Sun & Li, 2014):

$$L_0(\rho) = \sum_{j=1}^{M} \frac{\rho_j^2}{\rho_j^2 + \varepsilon} \tag{7}$$

where $\varepsilon$ is a focusing parameter. Application of $L_0(\rho)$ as stabilizer in minimization process of the
objective function (Eq. (3)) leads to the following choice of an $L_0$-norm constraint $W_{L_0}$ which is given
by (Last & Kubik, 1983):

$$[W_{L_0}]_j = ([\rho_j]^2 + \varepsilon)^{-1} \tag{8}$$

Based on Eq. (8) the $k^{th}$ iteration diagonal elements of the $L_0$-norm constraint matrix ($\boldsymbol{W}_{L_0}^k$) can be
formulated as follows:

$$[\boldsymbol{W}_{L_0}^k]_{jj}^{-1} = [\rho_j{}^{k-1}]^2 + \varepsilon \tag{9}$$

The focusing parameter $\varepsilon$ is a very important parameter. Its main purpose is to avoid singularities when
$\rho_j \to 0$. The parameter $\varepsilon$ is a small number and in general, we are interested in the case where $\varepsilon \to 0$
because a small value leads to very compact models. However, this may introduce instability. On the
other hand, if $\varepsilon$ is chosen large the $L_0$-norm compactness constraint has no influence on the compactness
of the model that means it results in a smooth solution. Figure 2 shows the comparison of the minimum
support stabilizing functional for different values of $\varepsilon$ to demonstrate the impact of the choice different
values of $\varepsilon$ further. From Fig. 2, one can see that as $\varepsilon$ becomes large the minimum support stabilizing
function loses its property and behaves more like the minimum length $L_2$-norm stabilizer which results





in undesirable smoothness in the model though it improves the stability. Therefore, it is very essential

to choose an optimal value of $\varepsilon$.

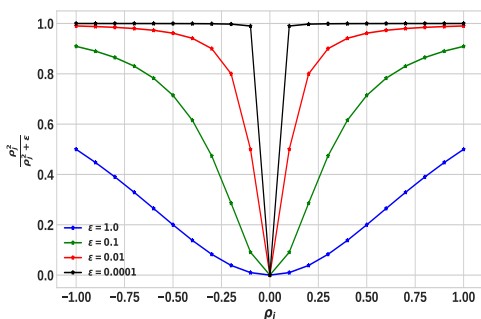

Figure 2: Comparison of the minimum support stabilizing function for different values of $\varepsilon$.


In previous investigations e.g Last & Kubik (1983) and Guillen & Menichetti (1984) the parameter $\varepsilon$
was assigned a value close to machine precision ($\approx 10^{-11} \quad to \quad 10^{-15}$). Alternatively, Zhdanov &
Tolstaya (2004) introduced a trade-off curve method, similar to the L-curve technique, to select $\varepsilon$ by
computing the model objective for the current model estimate over a range of values for $\varepsilon$. However,
as pointed out by Ajo-Franklin et al. (2007) setting $\varepsilon$ to values near machine precision results in severe
instability as $\rho_j \to 0$ and the approach of Zhdanov & Tolstaya (2004) often yields trade-off curves with
corners that are not well defined. Therefore it is better to fix $\varepsilon$ at a reasonable value determined by
experience, typically between $10^{-4} \quad to \quad 10^{-7}$ (Ajo-Franklin et al., 2007). Accordingly, in the present
work based on several numerical simulation tests, the value $10^{-6}$ is assigned just for the inversion
examples presented in the manuscript. Note that the developed method is flexible to use different values
of $\varepsilon$.

### 2.3.2 Error weighting

According to compact inversion method proposed by Last & Kubik (1983), the $k^{th}$ iteration error weight-
ing matrix $\boldsymbol{W_e^k}$ is defined as:

$$[\boldsymbol{W_e^k}]^{-1} = diag\left(\boldsymbol{A}[\boldsymbol{W_{L_0}^k}]^{-1}\boldsymbol{A}^T\right) \tag{10}$$

Even though $\boldsymbol{W_e^k}$ expressed by Eq. (10) is applied by many authors (Guillen & Menichetti, 1984;
Barbosa & Silva, 1994; Ghalehnoee et al., 2017; Gebre & Lewi, 2022), some instability was reported





by Lewi (1997, p. 87) in using $\boldsymbol{W}_e^k$ in scenarios such as complicated geological geometry and when the data is contaminated with noise. To overcome this problem Lewi (1997, p. 90) proposed to use a weighting matrix that make use of the following equation:

$$[\boldsymbol{W}_e^k]^{-1} = \left[\frac{[\sigma_\rho^2]^k}{1 + [\sigma_e^2]^k}\right] \boldsymbol{I} \tag{11}$$

where $\boldsymbol{I}$ represents identity matrix, and $\sigma_\rho^2$ and $\sigma_e^2$ are model and error variances respectively that are given by:

$$[\sigma_e^2]^k = \frac{\sum_{i=1}^{N}\{\boldsymbol{g_i} - \sum_{j=1}^{M} a_{ij}[\boldsymbol{\rho}_j^{k-1}]\}^2}{N - 1} \tag{12}$$

$$[\sigma_\rho^2]^k = \frac{\sum_{j=1}^{M}[\boldsymbol{\rho}_j^{k-1}]^2}{M - 1} \tag{13}$$

The term in square brackets in Eq. (11) can be considered as regularization parameter (Silva & Barbosa, 2006; Lewi, 1997, p. 90). Based on several numerical experiments done in the present work it was observed that this term can sometimes ends up in a larger value which may result over-regularization of the solution. For this reason, in the present study a new error weighting matrix $\boldsymbol{W}_{ne}^k$ is introduced and it is given as:

$$[\boldsymbol{W}_{ne}^k]^{-1} = diag\left(\boldsymbol{A}\left[\boldsymbol{W}_z(\frac{[\sigma_\rho^2]^k}{1 + [\sigma_e^2]^k})\boldsymbol{W}_h^k\right]\boldsymbol{A}^T\right) \tag{14}$$

Let us represent the terms in square brackets by $\boldsymbol{W}_n^k$ as follows:

$$\boldsymbol{W}_n^k = \boldsymbol{W}_z\left(\frac{[\sigma_\rho^2]^k}{1 + [\sigma_e^2]^k}\right)\boldsymbol{W}_h^k \tag{15}$$

where $\boldsymbol{W}_z$ and $\boldsymbol{W}_h^k$ are diagonal depth and hard constraint matrices respectively and will be described in the next subsections. Then the error weighting matrix in Eq. (14), the one introduced and implemented here becomes:

$$[\boldsymbol{W}_{ne}^k]^{-1} = diag\left(\boldsymbol{A}\boldsymbol{W}_n^k\boldsymbol{A}^T\right) \tag{16}$$

### 2.3.3 Auto-adaptive Regularization Parameter Estimation

Choosing a suitable value for regularization parameter is a crucial part of the inversion process. The precise value of regularization parameter depends on the noise level associated with the observed data. Thus, the higher value of $\ell$ refers to the higher noise level of the data points. Several methods have been proposed to choose the appropriate value of regularization parameters, and are reviewed in the literature



(Farquharson & Oldenburg, 2004; Vatankhah et al., 2014) and standard texts for example Vogel (2002, pp. 97-109) and Aster et al. (2018, p. 57). Particularly, depending on the noise level a constant value of $\ell$, throughout the inversion, has been chosen by many authors (Silva & Barbosa, 2006; Ghalehnoee et al., 2017). In other works, for example Zhdanov (2009) and Rezaie et al. (2017) the parameter $\ell$ has been iteratively updated in each iteration.

As pointed out in previous works (Farquharson & Oldenburg, 2004; Gholami & Aghamiry, 2017) instead of using a constant value of $\ell$, dynamic re-adjustment throughout the iterative scheme might be a superior approach. Taking this into account, in the present work $\ell$ is updated in each iterative step. In our implementation, to select an optimal regularization parameter at each iteration, we proposed an auto-adaptive regularization method. This method leads to an automatic update of the regularization parameter at each and every iteration. The basic principle including its procedure in relation to formally known adaptive regularization approach which was proposed by Zhdanov (2002, p. 55) and implemented by many authors (Zhdanov, 2009; Rezaie et al., 2017) is as follows. In adaptive regularization approach the initial value of the regularization parameter $\ell^1$ is updated at each iteration step by (Zhdanov, 2002, p. 55):

$$\ell^k = \ell^1 q^k \tag{17}$$

where $q$, as described by Zhdanov (2002, p. 55), is damping factor which decreases from iteration to iteration. Its initial value is empirically determined having a value between zero and one. It is obvious that the trial and error selection of the value for $q$ requires computational work . The presented auto-adaptive regularization method overcomes this problem and the iterative values $\ell^k$ are determined by the following formula:

$$\ell^k = \ell^{k-1} \left[ \frac{|\boldsymbol{g} - \boldsymbol{A}\boldsymbol{\rho}|_{max}^{k-1}}{|\boldsymbol{g} - \boldsymbol{A}\boldsymbol{\rho}|_{max}^{k}} \right] \tag{18}$$

where the term in the square bracket is an adjusting factor that is automatically determined at each iterative step. In the auto-adaptive regularization method, choosing a suitable initial value of $(\ell_o)$ is essential. Based on a number of synthetic and real data simulations done in this work we recommend the following in choosing a reasonable value of $\ell_o$: Firstly, the initial value of $\ell$ should be within the range $0 < \ell_o \leq 1$. Secondly the precise value of $\ell_o$ depends on the noise level related with the observed data. When the probable or expected noise level of the data is higher, a larger value $\ell_o$ is a reasonable choice to avoid unwanted and false anomalies due to noise. In contrast, when the probable or expected noise level is less a small value of $\ell_o$ should be chosen. Once an appropriate initial value $\ell_o$ is given as an





input, then for subsequent iterations Eq. (18) is used to determine $\ell^k$. The advantage of the auto-adaptive
regularization scheme is its capability to automatically determine a suitable regularization parameter, in
the course of the optimization process, depending on the automatically determined adjusting factor.

### 2.3.4  Physical Parameter Inequality Constraint

To produce a physically meaningful model from a gravity inverse solution, the usage of lower and upper
bound constraints on the recovered density contrast is beneficial (Silva et al., 2001; Grandis & Dahrin,
2014). Lower and upper bounds can be obtained from a prior information such as geological investiga-
tions in conjunction with published density values of rocks, well-logging, and/or laboratory tests. Many
procedures such as gradient projection approach (Wang & Ma, 2007; Lelièvre et al., 2009), transform
function approach (Pilkington, 2008) and logarithmic barrier approach (Li & Oldenburg, 2003) have
been applied in different inversion schemes to implement this constraint. However, with regard to $L_0$-
norm stabilizer based gravity inversion methods an effective method is the direct utilization of lower
and upper density constraint (Meng et al., 2018). Hence, in this work the direct density bound inequal-
ity constraint is used, that is at each iteration density contrast of each rectangular block is bounded by
minimum and maximum density constraint function given by:

$$
\begin{aligned}
&if \quad [\boldsymbol{\rho}^k]_j > [\boldsymbol{\rho}_{max}]_j \\
&if \quad [\boldsymbol{\rho}_{min}]_j < [\boldsymbol{\rho}^k]_j < [\boldsymbol{\rho}_{max}]_j \\
&if \quad [\boldsymbol{\rho}^k]_j < [\boldsymbol{\rho}_{min}]_j
\end{aligned}
\begin{cases}
[\boldsymbol{\rho}^k]_j = [\boldsymbol{\rho}_{max}]_j \\
[\boldsymbol{\rho}^k]_j = [\boldsymbol{\rho}^k]_j \\
[\boldsymbol{\rho}^k]_j = [\boldsymbol{\rho}_{min}]_j
\end{cases}
\tag{19}
$$

By using this function, if $k^{th}$ iteration $\rho_j$ of any block exceeds one of its bounds, then it will be fixed at
the violated bound.
In each iteration step the procedure to compute the hard constraint matrix $\boldsymbol{W}_h^k$ (Boulanger & Chouteau,
2001) and the reference density vector $\boldsymbol{\rho}_F^k$ is determined as follows: The diagonal elements of $\boldsymbol{W}_h^k$ are
fixed at $\varepsilon$ or 1.0. When a prior geological and geophysical information are able to provide the initial
value of density contrast of the $j^{th}$ specific cells, then these values are assigned to the corresponding
$[\boldsymbol{\rho}_F^k]_j$. Simultaneously, the corresponding diagonal elements of $[\boldsymbol{W}_h^k]_{jj}$ are set to be $\varepsilon$. During the in-
version process, if the $j^{th}$ elements of estimated density values falls out of inequality constraint limits
defined by $\rho_{min}$ and $\rho_{max}$, then $[\boldsymbol{\rho}_F^k]_j$ will be fixed at the violated bound density itself and $[\boldsymbol{W}_h^k]_{jj}$
assigned to be $\varepsilon$. On the other hand, if the elements of the estimated density did not exceed its bounds
(i.e. lies between the limits), $[\boldsymbol{W}_h^k]_{jj}$ and $[\boldsymbol{\rho}_F^k]_j$ are assigned to be 1.0 and 0.0 respectively.





Using $\boldsymbol{W}_h^k$ any blocks whose density is known from a priori information or exceeds the density con-
straint limit, the algorithm will automatically freezes this block in the next iteration by assigning a very
small weight to it. Whereas, $\boldsymbol{\rho}_F^k$ is used to remove the gravity effects of those cells that have crossed
the inequality constraint limit from the observed gravity data. That is applied to compute the reduced
gravity data vector $\boldsymbol{g}_r^k = \boldsymbol{g}^{obs} - \boldsymbol{A}\boldsymbol{\rho}_F^k$ in Eq. (4) of the inversion algorithm. In other word, at each
iterative step the inversion of subsequent iteration will be performed using reduced gravity data vector.

### 2.3.5 Depth weighting

It is well known that gravity data, like any potential field data, has no inherent depth resolution. The
reconstructed model structures by the inversion process tend to concentrate near the surface regardless of
the true depth of the causative bodies (Li & Oldenburg, 1996). This happens because the inverse solution
of model construction is a linear combination of kernel, whose amplitudes rapidly decay with depth. The
problem can be overcome by introducing a depth weighting matrix to counteract the natural decay of
kernel with depth (Li & Oldenburg, 1998). Depth weighting is designed to ensure that all cells have
equal likelihood to accommodate the sources, not just those at shallow levels that are most sensitive to
the observed data. Depth weighting is used and its effect is investigated by different authors (Pilkington,
2008; Commer, 2011). Based on Gebre & Lewi (2022), the recently proposed depth weighting function
is given as follows:

$$w_{zj} = (aZ_j + c_o)^{-\tau} \tag{20}$$

where $Z_j$ is the mean depth of the $j^{th}$ cell and $a$, $c_o$ and $\tau$ are adjustable parameters. The values of
the three adjustable parameters are computed by optimizing $w_z(z)$ to match with the actual gravity
kernel values utilizing nonlinear least-squares minimization (Virtanen et al., 2020). Accordingly, for all
inversions in this work the depth weighting matrix similar to the one used by Gebre & Lewi (2022) is
employed (Eq. (21)):

$$[\boldsymbol{W_z}]_{jj} = \boldsymbol{diag}(w_{zj}) \tag{21}$$

where $\boldsymbol{W}_z$ is diagonal M x M depth weighting matrix.

### 2.3.6 Stopping Criteria

It is clear that if the iterations are stopped too early, then a reasonable solution of the inverse problem may
not be obtained. On the other hand, too many iterations may waste computer time without increasing the





overall solution qualities. Thus, an important aspect of any iterative inversion method is to decide when
the iterations should be terminated. A number of stopping criteria have been proposed and employed to
terminate iterative inversion algorithms (Borges et al., 2015; Levin & Meltzer, 2017). Commonly used
stopping criteria are based on a norm of the residual vector (i.e. the norm of the difference between
estimated and observed data). For instance, a noise level, i.e. $\chi^2 = ||\boldsymbol{W}_d(\boldsymbol{g}^{obs} - \boldsymbol{A}\boldsymbol{\rho})||_2^2$, where a
diagonal data weighting matrix $\boldsymbol{W}_d$, whose $i^{th}$ element is the inverse of the standard deviation of the
noise at each data point, is used by Boulanger & Chouteau (2001) and Vatankhah et al. (2017). Other
criteria for stopping gravity inversion procedure are based on simple $misfit$ or the Root Mean Square
Error ($RMSE$) between the observed data and predicted data produced by the recovered model (see, for
example Rezaie & Moazam (2017)). The expressions used to estimate these criteria are the following:

$$misfit = \left( \frac{\sum_{i=1}^{N}(\boldsymbol{g}_i^{obs} - \boldsymbol{g}_i^{cal})^2}{\sum_{i=1}^{N}(\boldsymbol{g}_i^{obs})^2} \right)^{\frac{1}{2}} \tag{22}$$

$$RMSE = \frac{(\sum_{i=1}^{N}(\boldsymbol{g}_i^{obs} - \boldsymbol{g}_i^{cal})^2)^{\frac{1}{2}}}{N} \tag{23}$$

Ekinci (2008) also introduced other possible criterion, namely the parameter variation function ($smy$)
which is defined as:

$$smy = \left( \sum_{j=1}^{M}(\boldsymbol{\rho}_j^{k} - \boldsymbol{\rho}_j^{k-1})^2 \right)^{\frac{1}{2}} \tag{24}$$

The most widely used approach is to quit the iterative process when one of the above criteria are below
a given tolerance (the level of observational error). However, in practical applications a precise value
for such tolerance is rarely known; rather, only some possibly vague idea of the desired quality of
the numerical approximation is at hand. Moreover, it has been pointed out by Rao et al. (2018) that
stopping iteration based solely on the norm of the residual is neither safe nor a robust solution. The
non-uniqueness and instability of the gravity inverse problem further complicates the usage of only
one of the aforementioned stopping criteria. To overcome these issues, a combination of the $misfit$
and $smy$ has been utilized in this paper. Therefore, the iterative procedure continues until one of the
following stopping criteria is met: (I) the maximum number of iteration ($k_{max}$) given by the user is
reached or (II) the difference between two consecutive iteration values of $smy$ and $misfit$ have reached
the target values. That means for the second criterion both the conditions $|smy^{k-1} - smy^k| \leq \tau$ and
$|misfit^{k-1} - misfit^k| \leq \mu$ must be satisfied at the same time. In all demonstrations considered in this





work, after testing different values, the parameter $\tau$ is assigned to $\sqrt{2M}$; and $\mu$ to 0.005. Where M is again the total number of model parameter. The effectiveness of the proposed termination criteria will be illustrated by using synthetic tests.

## 2.4 Computational procedure

The solution of the linear system of equations in Eq. (2) will be carried iteratively using the information about the misfit and density from successive iteration. The input parameters for the inversion procedure are: (1) Kernel matrix ($\boldsymbol{A}$) and discretized subsurface model (mesh) and its initial approximation reference density model $\boldsymbol{\rho}_F$ if exits based on a priori information; (2) Observed gravity anomaly ($\boldsymbol{g}$) at measurement points ($\mathbf{x}$); (3) Maximum number of iteration ($k_{max}$) and the constant $\beta$; (4) Lower $\boldsymbol{\rho}_{min}$ and upper $\boldsymbol{\rho}_{max}$ density bounds and initial $\ell_o$ value. In summary, the steps taken to carry out the inversion process consists the followings.

1. For k = 0, if there is no a priori information, $\boldsymbol{W}_{L_0}$, $\boldsymbol{W}_c$, $\boldsymbol{W}_n$ and $\boldsymbol{W}_h$ are identity matrices, $\boldsymbol{\rho}_F = 0$. $\boldsymbol{W}_z$ and $\boldsymbol{W}_{ne}$ are computed through Eq. (21) and (16) respectively, after this, the first iteration model parameter solution is obtained by Eq. (4).

2. The elements of $\boldsymbol{W}_h$ and $\boldsymbol{\rho}_F$ are updated as explained in preceding section, then $\boldsymbol{W}_{L_0}$ is calculated using Eq. (9) and then $\boldsymbol{W}_c$ using Eq. (5).

3. Compute the value of $\sigma_\rho$ and $\sigma_e$ using expressions (13) and (12) respectively. Then calculate $\boldsymbol{W}_n$ using Eq. (15).

4. To remove the effect of those blocks that have crossed the maximum target density, evaluate the reduced data $\boldsymbol{g}_r^k = \boldsymbol{g}^{obs} - \boldsymbol{A}\boldsymbol{\rho}_F^k$. Then compute the current $\ell$ with Eq. (18) and $\boldsymbol{W}_{ne}$ with Eq. (16).

5. Carrying out the inversion through Eq. (4).

6. Application of bounded constraints on density are carried out as discussed in the preceding section.

7. Now a forward modelling procedure will be carried out using Eq. (2) to compute the gravity anomaly $\boldsymbol{g}^{cal}$ from the estimated model in the previous iteration.





8. Data $misfit$ (Eq. (22)) and $smy$ (Eq. (24)) are computed using $g^{cal}$ from step 7, and obtained model parameters from the previous and current iteration.

9. Test if the stopping criteria are fulfilled. If the termination criteria are satisfied the iteration terminates and obtained results are stored and plotted. Otherwise, using the current estimated density model, move to the next iteration k by going to the second step and continue the iterative procedure until the stopping criteria are fulfilled.

# 3   Synthetic Model Test

To evaluate the functionality and efficiency of the method, the developed procedure was tested on several synthetic model examples. The examples presented here are randomly chosen to demonstrate: (I) the applicability of the proposed auto-adaptive regularization technique (Eq. (18)) and error weighting function (Eq. (16)); (II) the performance of the method in producing compact and sharp images of the causative bodies; (III) the effectiveness of the combined stopping criterion. The forward and the inverse problem were carried out using the procedure described in the preceding sections. In the inversion of the synthetic examples, the same subsurface discretization as the one used in generating the synthetic data (Forward modeling) is used. All the inversion tests are performed on a Desktop computer ( $11^{th}$ Gen Intel(R) Core(TM) i7-11700, 2.50GHz processor). For the first and second synthetic examples presented in this work: (I) The model region was discretized into 60 x 15 rectangular cells and the dimensions of each cell were taken as 10 x 10 m, in the X and Y directions respectively. (II) The synthetic gravity data were computed at 60 data points that are centered in each cell at the top side of the model, to produce data at 10 m sample interval. (III) The computed gravity anomalies are contaminated with Gaussian noise that has a standard deviation that amounts to 4 % of the magnitude at each data point with zero mean (Farquharson, 2008; Rezaie et al., 2017).

The first synthetic data inversion has been done for the model presented in Fig. 3(a). For this synthetic model the causative bodies are two rectangular structures elongated differently in the horizontal and vertical directions and located at different depths. The causative bodies have the same density contrast $1000\ kg/m^3$. The density of the causative bodies are given relative to the zero density of uniform background. Figure 3(a) upper panel shows noise free (solid line) and noise contaminated (star dots) gravity data. Separate inversion runs, for three different $\ell_o$ values ( 0.2, 0.3 and 0.4 ), were performed with the developed inversion method. Note that, for subsequent iterations the proposed auto-adaptive regular-



 ization technique (Eq. (18)) is used to compute $\ell$ for each case. At the beginning of the inversion, the

iterations are initialized with $\boldsymbol{\rho}_F = 0$ and $\boldsymbol{W}_h = \boldsymbol{W}_c = \boldsymbol{W}_n = \boldsymbol{W}_{L_0} = \boldsymbol{I}$. The lower limit density

contrasts of all cells is zero ($\boldsymbol{\rho}_{min} = 0$) and the upper bound $\boldsymbol{\rho}_{max} = 1000\ kg/m^3$.

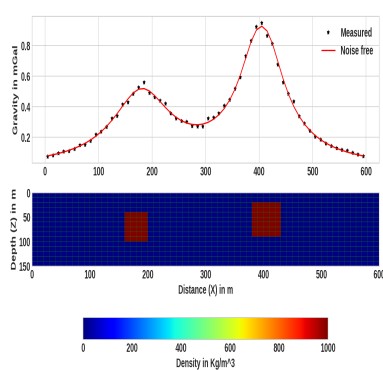
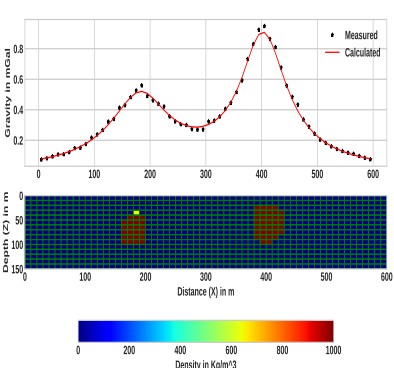

(a) The lower panel represents 2-D synthetic model, which constitutes two isolated rectangular bodies located at various depths and the top panel shows the gravity anomaly due to these two subsurface rectangular bodies.

(b) The lower panel represents the subsurface, as a result of the proposed inversion method using $\ell_o = 0.3$ and the top panel shows the synthetic data together data derived from the model.

Figure 3: The first synthetic model and the result of the inversion.


The results of the inversion by using the developed method for three different $\ell_o$ values are shown in
Figs. 3(b) and 4. The corresponding data fit between the predicted gravity anomaly (solid line ) and
actual contaminated data (stars) are also shown. Comparing the inversion results with the original syn-
thetic model in Fig. 3(a), the inversion has sufficiently recovered the true models. The depth, geometry,
and density distributions of the synthetic causative bodies were recovered adequately. This can confirm
the applicability of the proposed auto-adaptive regularization technique (Eq. (18)) and error weighting
function (Eq. (16)). Notice that the results also indicate the robustness and stability of the developed
inversion method for different $\ell_o$ values. The avarge computation time to finish the inversion is approx-
imately 16.3 seconds.
The second synthetic model is more complicated and consists of two causative bodies placed at various
depth. The bodies have different sizes, shapes, and density contrasts. The first causative body is a verti-
cal rectangular block, with density contrast 2000 $kg/m^3$, placed at 40 m depth and the second body is
a dipping dike with density contrast 3000 $kg/m^3$ at 20 m depth. The synthetic model is shown in the
lower part of Fig. 5(a) and the generated noise-corrupted and noise free gravity data are shown on the





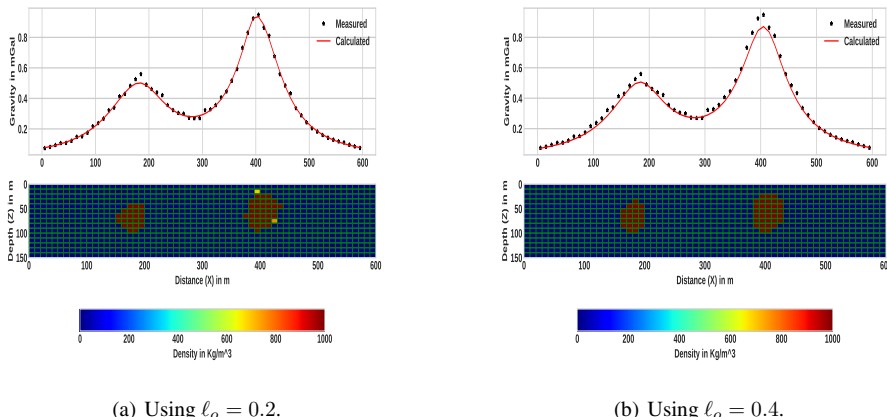

(a) Using $\ell_o = 0.2$.          (b) Using $\ell_o = 0.4$.

Figure 4: Inversion results, using different $\ell_o$ values, for the first synthetic model given in Fig. 3(a).

upper part. Using the generated synthetic data, the inversion was initiated by assigning an initial zero
density to each cell. We set initial $\ell_o = 0.3$. The density contrast limits are bounded between lower
bound $\boldsymbol{\rho}_{min} = 0$ and the upper bound $\boldsymbol{\rho}_{max} = 3000\ kg/m^3$. Even though a maximum iteration of 20
was set, the $misfit$ and $smy$ between two consecutive iterations gradually fall below the threshold set
after the $14^{th}$ iteration. The total computation time is approximately 15.73 seconds. In Fig. 5(b), the
resulting model from the inversion of the second synthetic model (Fig. 5(a)) using the proposed method
is presented. As can be seen in Fig. 5(b) upper panel the modeled gravity data (solid line) fits adequately
with the synthetic data. The result, presented in Fig. 5(b) lower panel, indicates an acceptable recon-
struction of the synthetic multi-sources and multi-shape bodies that are located at different depths. The
true shape, location and density of the causative bodies are recovered adequately. Like the first example
the reproduced images of the localized multiple sources are compact and sharp (Fig. 5(b) lower panel).
For the third and fourth synthetic examples: (I) The subsurface model was discretized into 100 x 20
rectangular cells. Each cell has a size of 50 m in X and Z directions. (II) The synthetic gravity data
were computed on 100 data points with a sample spacing of 50 m. The third synthetic model includes
two dipping dikes in opposite directions. The causative 2-D bodies have different sizes and the same
density contrast that amounts to $1000\ kg/m^3$ in a homogeneous background zero density. The top part
of the shallower dipping dike lies at a depth of 200 m and that of the deeper dike at a depth of 250
m. The computed gravity anomalies were contaminated by uncorrelated Gaussian noise whose standard
deviation was equal to 4% of the difference between the maximum and the minimum anomaly and zero





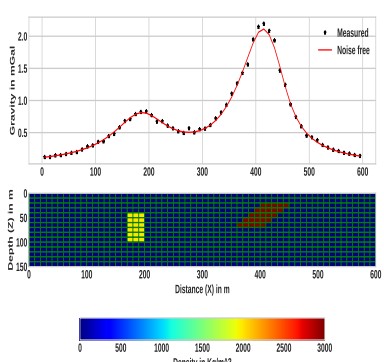
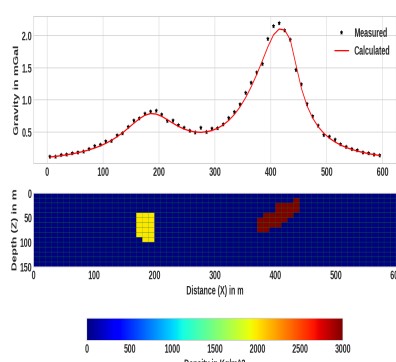

(a) Synthetic model consisting of a dipping dike and vertical rectangular block and the corresponding gravity data.

(b) The density model obtained by inverting the gravity data using the developed method. The predicted data as a result of inversion process are shown on the top panels (solid line).

Figure 5: The second example synthetic model and the corresponding inversion result.

mean. The synthetic model and the corresponding data are shown in Fig. 6 at lower and upper panels

respectively.

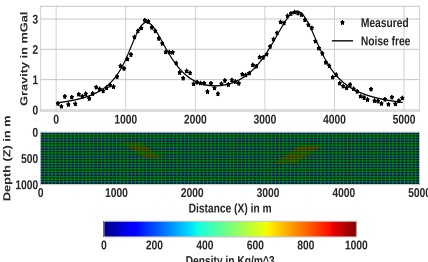

Figure 6: The third synthetic model that comprises two dikes at various depths with the density contrast that amounts to 1000 $kg/m^3$ and the corresponding gravity data.


The inversion process was commenced by setting the densities of all cells to zero. The initial value of
$\ell_o$ was set to 0.4. The bounding density ranges were set to a minimum value $\boldsymbol{\rho}_{min} = 0$ and maximum
value $\boldsymbol{\rho}_{max} = 1000\ kg/m^3$. The maximum number of iterations was set to 20. Here, the inversion
converged after the $13^{th}$ iteration and the total computation time is approximately 66.49 seconds. The
resulting model and the inverted data using the proposed method are shown in Fig. 7(b). For the sake
of comparison keeping all inversion parameters the same, the synthetic data was also inverted with the
classical $L_2$-norm regularized inversion approach and the obtained result is shown in Fig. 7(a). As it




can be seen from the lower panel of Fig. 7(b), unlike the model in Fig. 7(a), the developed method was
able to produce a compact and sharp model successfully. The other concern, which can be seen from
the result in Fig. 7(a), is that the target density contrast values are underestimated in the case of the
conventional $L_2$-norm inversion. In contrast, the geometrics, locations, and densities of both anomalous

structures were adequately recovered with the presented inversion method (see Fig. 7(b)).

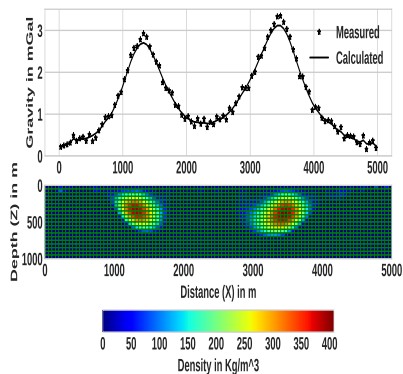
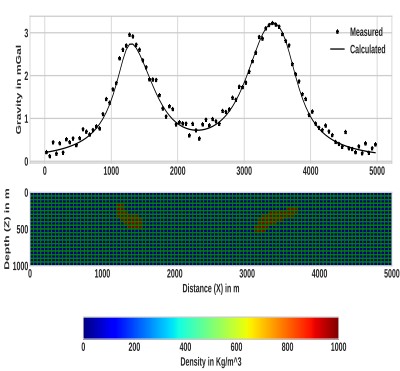

(a) Using the conventional minimum norm ($L_2$-norm) smooth stabilizer and the corresponding data fit.

(b) Using the presented method

Figure 7: Inversion results of the third synthetic example in Fig. 6.


The fourth synthetic model consists of two different rectangular anomalous bodies (Fig. 8(a) lower
panel). The anomalous structures have different dimensions and are buried at different depths. The top
of the first rectangular block is placed at a depth of 200 m and its density contrast is -1000 $kg/m^3$ while
the top of the second block is placed at a depth of 250 m and has a density contrast of 1000 $kg/m^3$.
Different density contrast, size, and depth of adjacent structures have been considered to show the ability
of the presented inversion method in reconstructing true parameters for these models. In this synthetic
example, the computed anomalies are contaminated by Gaussian noise with a standard deviation of 3%
of the difference between the maximum and the minimum anomaly.
For the current example, the inversion process was initialized by setting the initial value of $\ell_o$ = 0.5.
The lower bound for the density constraint $\rho_{min}$ = -1000 $kg/m^3$ and the upper bound $\rho_{min}$ = 1000
$kg/m^3$. Similar to the previous examples, though the maximum number of iterations was set to be 20,
the iterative step terminated when the proposed combined criterion is satisfied after 11 iterations. The ap-
proximate running time required to finish the inversion is 55.64 seconds. Figure 8(b) lower panel shows
the recovered density contrast model. The corresponding fits between synthetic (stars) and predicted



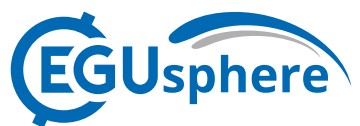

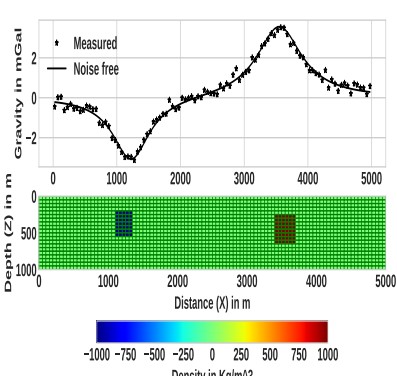 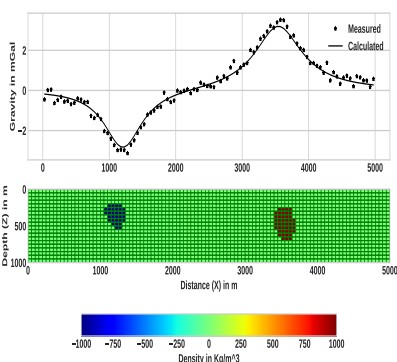

(a) Synthetic model consisting of two rectangular bodies at various depths with different density contrast and the corresponding noise free and contaminated gravity anomalies.

(b) The lower panel shows recovered density contrast model obtained by inverting the gravity data using the developed method, while the upper one shows the associated fits between the synthetic data that is taken from (a) and the predicted response.

Figure 8: The fourth synthetic model example and the corresponding inversion result.

data (line) are shown in the upper panel of the same figure. We can see that the recovered rectangular
bodies are compact and have sharp boundaries. The obtained results also indicate that the depth and
density contrast of the anomalous rectangular bodies have been determined sufficiently.
Here, the effectiveness and the advantage of the proposed combined stopping criterion are illustrated
by comparing it with another commonly used stopping condition. For this reason, the inversion process
was performed again with the developed inversion method using only the misfit function ($|misfit^{k-1} -$
$misfit^{k}|$) as a stopping condition. Note that, for comparison purposes, all the other inversion parame-
ters are set the same except for the stopping criterion. The resulting recovered density contrast models
and the data fit are presented in Fig. 9. The corresponding values of the $misfit$ and $smy$ as a function
of iteration number are also shown in Fig. 10(a). For the sake of comparison, the $misfit$ and $smy$
when using the proposed combined stopping criterion for the same data set are also presented in Fig.
10(b). The stopping condition $|misfit^{k-1} - misfit^{k}| \leq \mu$ was reached after 5 iterations, as shown
in the curve of Fig. 10(a) before the true density distribution has been recovered fully. In other words,
the estimated models are not satisfactory because densities lower than the target density are observed
around the edges of the anomalous bodies ( Fig. 9). This indicates that unlike the result presented in
Fig. 8(b), where the proposed combined stopping condition is used, quitting the iterative process only
with $|misfit^{k-1} - misfit^{k}| \leq \mu$ criterion produces a premature solution that is before the maximum




compactness is achieved.

A number of other numerical experiments we carried out showed that there are situation where either

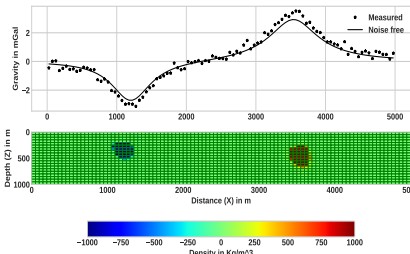

Figure 9: Inversion result obtained using only the commonly used criterion ($|misfit^{k-1} - misfit^k|$) and the corresponding data fit (upper panels) for the synthetic example in Fig. 8(a). The obtained density model shows that compact and sharp model is not approximately achieved due to the termination before the iterative procedure has reached convergence.

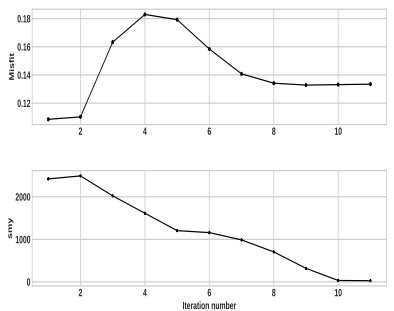
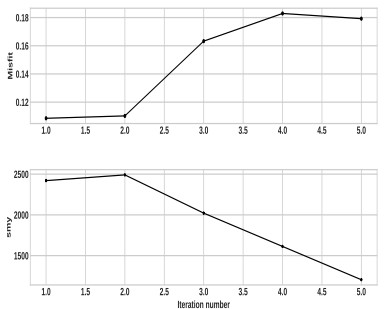

(a) Using the proposed combined stopping condition    (b) Using only $|misfit^{k-1} - misfit^k| \leq \mu$

Figure 10: The progression of $misfit$ and $smy$ in the course of the iteration during the inversion of the fourth example synthetic data.


$misfit^k|$ or $|misfit^{k-1} - misfit^k|$ fall below the given threshold values, at earlier iterations, before the
true density is fully recovered. Thus, it is hard to take only one criterion as a termination condition. As
stated in Sect. 2.3.6, it has been mentioned that the same has also be pointed out in number of previous
works (Rao et al., 2018). Whereas, in the case of the proposed criterion that is when both the conditions
$|smy^{k-1} - smy^k| \leq \tau$ and $|misfit^{k-1} - misfit^k| \leq \mu$ are satisfied at the same time the inversion
process yields an acceptable model. This clearly illustrates the advantage of using the proposed stopping
criterion and its effectiveness in quitting the iterative scheme after optimal number of iterations. To fur-
ther illustrate the effectiveness of the proposed combined criterion, the inversion process is allowed to




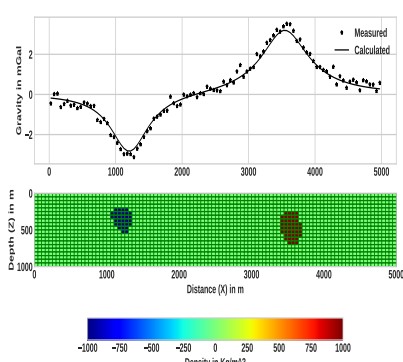
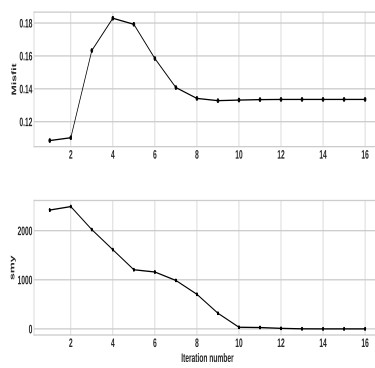

(a) The obtained recovered density model (lower panel) and the corresponding data fit (upper panel).

(b) Progression of $misfit$ (top panel) and $smy$ (lower panel) in the course of the iterative procedure.

Figure 11: Late iteration termination (at $16^{th}$ iteration) inversion result and the corresponding $misfit$ and $smy$ variation with iteration number for the fourth example in Fig 8.

continue to the $16^{th}$ iteration and the model as a result of this is presented in Fig. 11(a). The progression
of the $misfit$ and $smy$ in the course of the iterative procedure are also given in Fig. 11(b). As it can be
seen from the result (Fig. 11(b)) the solution obtained at subsequent iterations, after the $11^{th}$ iteration
where the iteration is terminated with the proposed stopping condition, remains virtually unaltered. This
can also be observed from the $misfit$ and $smy$ variation curves shown in Fig. 11(b), in such that after
$11^{th}$ iteration the $misfit$ and $smy$ values remain literally unchanged. Moreover, the results also indicate
the appropriateness of the suggested threshold values $\mu$ and $\tau$ used in the proposed stopping criterion.
The other thing one can observe from the results in Fig. 11 is the stability of the developed inversion
method. This can also illustrate the effectiveness of the newly proposed auto-adaptive regularization
technique (Eq. (18)) and error weighting function (Eq. (16).
In general, the presented method was tested with noise contaminated data that are generated from dif-
ferent geometries, locations, sizes, and densities contrasts of causative bodies and it has successfully
recovered all models. Moreover, all the reconstructed images of the presented synthetic models are
compact and sharp. Numerous synthetic data inversions were performed to analyze the impact of the
density contrast bounds. The obtained results, which are not presented here, suggest that the values of
density contrast bounds have a significant effect on the results, and hence to recover a feasible model a
good knowledge of the density bounds is vital. This also pointed out by number of authors, for exam-
ple Vatankhah et al. (2017); Li et al. (2018) and Utsugi (2019), in the case of inversion methods that



<sup></sup>

use non-smooth stabilizers ($L_1$-norm or $L_0$-norm). Provided that the lower and upper density contrast
bounds are chosen properly, this inversion technique produces acceptable solutions. Therefore, as it was
demonstrated using synthetic examples, the proposed method has effectively and efficiently recovered
the synthetic models. Generally, the tests performed on different geometry synthetic models showed that
the method gives acceptable results for localized multi-sources anomalies at different depths with sharp
features.

## 4  Real Data Test

To test the method in the real world, where the gravity data is contaminated with noise the improved
algorithm is implemented on gravity data acquired on different published geologic settings. The first one
is taken from Green (1975) by carefully digitizing the residual gravity data. As it was given in Green
(1975) the data was measured over the Guichon Creek batholith in south-central British Columbia. For
the details about the measurement and geology the reader is referred to Ager et al. (1973) and Ager
(1972). The residual gravity profile is digitized at a regular intervals of 0.5 km to produce a total of 64

data points as shown in Fig. 12. For the inversion, the source volume beneath the anomaly was divided

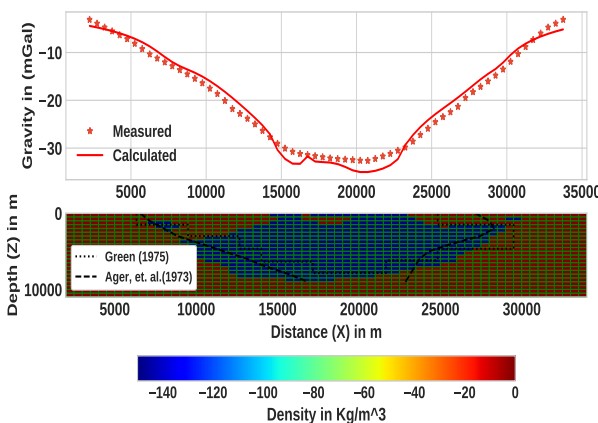

Figure 12: The observed gravity anomaly over Guichon Creek batholith in south-central British Columbia (after Green (1975)) and its inversion result. Digitized data (star marks) with calculated data (solid line) shown on the top panel. Corresponding recovered density contrast model after 9[th] iteration shown on the bottom panel. The recovered body density contrast is represented by the color scale bar. For comparison, the results obtained by Ager et al. (1973), which was obtained from drilling and Green (1975) are also presented.






into 64 x 22 square lattice with dimensions of each cell being 0.5 km in both X and Z-directions. Based

on the a prior information from Ager (1972) density values were constrained between the limits $\rho_{min}$

= -150 $kg/m^3$ and $\rho_{max}$ = 0.001 $kg/m^3$. We start the inversion with a homogeneous initial model in

which every block has the same zero density and an initial $\ell_o$ value of 0.48. The inversion was terminated

after $9^{th}$ iteration because the stopping criteria are fulfilled. The resulting model is presented in Fig. 12.

For comparison, the results obtained by Ager et al. (1973) and Green (1975) are also included in Fig.

12. The shape, real extent of the anomaly, and depth to bottom from the developed method are very

close to the true geological feature (Ager et al., 1973) which was obtained from drilling. That means the

implementation of the presented method resulted in a better solution compared Green (1975). Note that,

this reasonable result is obtained by using only the density contrast limits as a prior information.

The second test on measured gravity data is carried out using the published data by Last & Kubik

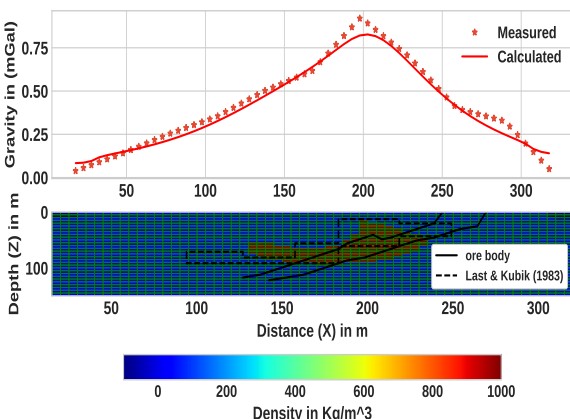

Figure 13: An observed gravity anomaly over the Woodlawn ore body, New South Wales (After Last & Kubik (1983)) and the its inversion result. The digitized data (star marks) is shown together with calculated data (solid line) on the top panel. The corresponding recovered density contrast model after $11^{th}$ iteration shown on bottom panel and the ore body proved by drilling is shown with solid line. The recovered body density contrast is represented by the color scale bar.

(1983) over the Woodlawn massive sulfide ore body, New South Wales, Australia. The residual anomaly

of the area consisting of 61 data measurements, sampled every 5 m, is digitized from Last & Kubik

(1983). The details about the data measurement and the geology of the area are discussed in Whiteley

(1981). The model subsurface was divided into 61 by 30 blocks with a dimension of 5 m in both X- and

Z-direction. Inverse modeling was performed with bounding constraints $\rho_{min} = -600$ and $\rho_{max} = 1000$



$kg/m^3$. The initial given value for $\ell_o$ is 0.6. The final solution was obtained after the 11$^{th}$ iteration. The
reconstructed model including the final model of Last & Kubik (1983) are shown in Fig. 13. The cross-
section of the ore body verified by drilling (Whiteley, 1981) is also shown in the figure. The recovered
model is approximately coincident with the shape, depth of burial and density of the known ore body.
Areas of misfits in the current and previous works are believed to be caused by the termination of the
original data at both ends before it reaches the background level. Thus, this can be additional evidence
that the presented method can be successfully applied to real data.

## 5   Conclusion

We have presented an alternative gravity inversion method that can produce compact and sharp images
by using the L$_0$-norm stabilizing functional that helps to model geological features with non-smooth,
blocky geologic bodies. Physical parameter inequality constraints, and depth weighting are integrated
into the procedure. The method also incorporates an auto-adaptive regularization technique, which auto-
matically determines a suitable regularization parameter at every iteration, and an error weighting func-
tion that helps to improve both the stability and convergence of the method. One of the strongest sides of
the proposed auto-adaptive regularization and error weighting matrix is that they are not dependent on a
priori knowledge of the noise level. Because of that, the method can yield reasonable results even when
the noise level of the data is not known properly. We implemented a combined stopping criteria and
illustrated its effectiveness to terminate the iterative inversion process after an optimal number of steps.
To illustrate the efficiency and the capacity of the proposed procedure numerous synthetic tests were
done. From these, four synthetic examples were presented. According to the results from these syn-
thetic examples, the method can be applied for multi-source localized bodies located at different depths
and having different geometries with sharp features. Furthermore, the method proved to be efficient in
resolving causative bodies both vertically and laterally and produced compact and sharp images. The
obtained results also indicate that the method behaves well with different noise levels embedded in the
data and still retains its stability. This can confirm the robustness and stability of the developed inversion
method for different noise levels. The method was also tested on measured gravity data. We obtained
geologically acceptable models and the results showed that our approach is effective and reliable. From
a computational point of view, the method is efficient and can be easily run on a personal computer just
in a few seconds. In conclusion, the developed method is advantageous in such that it is stable, efficient,



and resolves sharp subsurface futures with acceptable resolving capacity.



# Data Availability

The authors confirm that the real data supporting the findings of this study are available within the articles:

1. Green, W. R. (1975) Inversion of gravity profiles by use of a backus-gilbert approach. Geophysics, 40, 763–772 . and its supplementary material.

2. Last, B. and Kubik, K. (1983) Compact gravity inversion. Geophysics, 48, 713–721. and its supplementary material

# Author contributions

MGG developed the methodology; EL supervised the research work; MGG wrote the manuscript draft; EL reviewed and edited the manuscript.

# Competing interests

The authors declare that they have no known competing financial interests or personal relationships that could have appeared to influence the work reported in this paper.

# Acknowledgements

This work was sponsored by Wolkite and Addis Ababa Universities. We are thank full to all members of the Institute of Geophysics, Space Science and Astronomy of Addis Ababa University for all their assistance and allowing to use different office and computational facilities. Most importantly we thank Filagot Mengistu for her limitless support to this research work.



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





## Figure Captions

Figure 1. A 2-D model of the subsurface under a gravity profile. Gravity stations ($X_i$) are located at the centers of the blocks, indicated by the $\bigtriangledown$ symbols.

Figure 2. Comparison of the minimum support stabilizing function for different values of $\varepsilon$.

Figure 3. The first synthetic model and the result of the inversion.

Figure 4. Inversion results, using different $\ell_o$ values, for the first synthetic model given in Fig. 3(a).

Figure 5. The second example synthetic model and the corresponding inversion result.

Figure 6. The third synthetic model that comprises two dikes at various depths with the density contrast that amounts to 1000 $kg/m^3$ and the corresponding gravity data

Figure 7. Inversion results of the third synthetic example in Fig. 6.

Figure 8. The fourth synthetic model example and the corresponding inversion result.

Figure 9. Inversion result obtained using only the commonly used criterion ($|misfit^{k-1} - misfit^k|$) and the corresponding data fit (upper panels) for the synthetic example in Fig. 8(a). The obtained density model shows that compact and sharp model is not approximately achieved due to the termination before the iterative procedure has reached convergence.

Figure 10. The progression of $misfit$ and $smy$ in the course of the iteration during the inversion of the fourth example synthetic data.

Figure 11. Late iteration termination (at $16^{th}$ iteration) inversion result and the corresponding $misfit$ and $smy$ variation with iteration number for the fourth example in Fig. 8(a).

Figure 12. The observed gravity anomaly over Guichon Creek batholith in south-central British Columbia (after Green 1975) and its inversion result. Digitized data (star marks) with calculated data (solid line) shown on the top panel. Corresponding recovered density contrast model after $9^{th}$ iteration shown on the bottom panel. The recovered body density contrast is represented by the color scale bar. For comparison, the results obtained by Ager (1973), which was obtained from drilling and Green (1975) are also presented.





Figure 13. An observed gravity anomaly over the Woodlawn ore body, New South Wales (After Last and Kubik (1983)) and the its inversion result. The digitized data (star marks) is shown together with calculated data (solid line) on the top panel. The corresponding recovered density contrast model after 11[th] iteration shown on bottom panel and the ore body proved by drilling is shown with solid line. The recovered body density contrast is represented by the color scale bar