# Peer review of "Gravity Inversion Method to Produce Compact and"

_EGUsphere, 2022_

## Referee Comment (RC2)

This work contains interesting improvements over a previous work from the authors (Gebre and Lewi, 2022) concerning gravity inversion using L0-norm regularization. The main contributions include auto-adaptive regularization and combined stopping criteria. The results shown include many tests with synthetic data and real data which supports the claim of the article. In general, the article is well written and should be accepted after few minor modifications.

- The article contains typos and minor grammar mistakes that should be corrected. For example, in line 122, it should be "Here the misfit functional is $\Phi_d$=... and $\mathbf{W}_e$ is the error...". In line 334, it should be "total number of model parameters", etc...

- For the sake of completeness, the expression of the stabilizing functional, $S(\rho)$ should be given to explicit the role of $\mathbf{W}_c^k$ in the objective function and improve the understandability of (4).

- Also, as the method is based on iteratively reweighted least squares (IRLS), it should be mentioned on the text.

- In (18), the max operation should be defined.

- In (19), please use a notation similar to:

$$[\tilde{\rho}^k]_j = \begin{cases} [\rho_{max}]_j \ if \ [\rho^k]_j > [\rho_{max}]_j \\ [\rho^k]_j \ if \ [\rho_{min}]_j \leq [\rho^k]_j \leq [\rho_{max}]_j \\ [\rho_{min}]_j \ if \ [\rho^k]_j < [\rho_{min}]_j \end{cases}$$

---

## Author Comment (AC1)

**Reviewer 1: RC1– Anonymous**

| Reviewer | Comments/corrections /Questions | Accepted / not Accepted | Authors Response |
|---|---|---|---|
| RC1 | **General Comments:** The authors present a sparsity-constrained inversion method. The technical content of the paper is good and have both synthetic and field data illustrations. However, the paper has several typo and grammatical errors. The following are my comments on the paper. | Accepted | *First of all, we would like to thank **Reviewer RC1** for the careful revision of the manuscript. We highly appreciated his questions and valuable comments. **Note:** We have accommodated nearly all the suggestions as they were very important. The applied changes are highlighted in yellow color in the revised manuscript.* |
| RC1 | **Comments//Questions 1:** I suggest the title be shortened to **"Gravity Inversion Method Using L0-norm Constraint with Auto-adaptive Regularization and Combined Stopping Criteria"** | Accepted | *We have shortened the title as suggested.* |
| RC1 | **Comments//Questions 2:** Could you discuss the possibility of extending the method to 3D? | Accepted | *According to the reviewer's suggestion, we have incorporated a text about the possibility of extending the method to a 3D inversion algorithm.* |
| RC1 | **Comments//Questions 3:** For the field data examples, can you show the conventional least square inversion results like the one shown in Fig. 7a. | Accepted | *Because it was extensively discussed in previous work and because we showed the same using the synthetic data we were more focused on showing the advantage of the* |

| | | | |
|---|---|---|---|
| | | | *new approach compared to other previous work. However, because it will add value to the manuscript we have included the least-square solution for one of the field data as suggested by Reviewer RC1, for better justification and clarification.* |
| RC1 | **Comments//Questions 4**: For the synthetic data examples, is the noise added in the gravity data or the model density? The description in the paper is not clear about this point. | Accepted | *For all presented* synthetic data examples *the noise is added in the gravity data as mentioned in the text. To make this point clearer we have rewritten additional* descriptions *in the revised manuscript.* |
| RC1 | **Comments//Questions 5:** The noise added in the synthetic data is small. Can you show the robustness of the method by adding significant of noise in the data? | Not Accepted | *The noise added to the synthetic data is comparatively larger than most of the previously published works and it took into consideration the error budget in measuring gravity data presently. That is commonly considers the real data scenario. Different inversion methods have been published using different approaches for adding Gaussian noise. As an example, the following works used different ways for adding the Gaussian noise: Li and Oldenburg (1998); Boulanger and Chouteau (2001); Cella* |

| | | | |
|---|---|---|---|
| | | | *and Fedi (2012); Vatankhah et al. (2014). For the first two examples, we have used a similar computation scheme applied by several researchers e.g. Li and Oldenburg 1998 (used 2%); Farquharson, 2008 ( used 1 %); Portniaguine and Zhdanov, 2002 ( used 2 % ); Rezaie et al., 2017 (used 3 %). **Note** Please note that we have used 4 %. To show the robustness of the presented method further, for the third and fourth examples we used another computation scheme of the noise which is even more strong as we can clearly see from the presented Figures in the manuscript.* |
| **RC1** | **Comments//Questions 6:** What happens when the causative body is big in size but has a sharp boundary? | Accepted | *The developed method can successfully recover a causative body which is big in size, with a sharp boundary. This is because the method uses one of the well-known sparse norm constraints which is used to recover non-smooth or blocky geological features. For example, Feng et. al 2020 applied a similar L0 norm constraint to estimate the basement relief of a rift basin consisting of grabens and horsts. Moreover, the* |

|  |  | *capability of the presented method* can be demonstrated by the first real data example in the manuscript where the geological structure is *b*ig and also has a sharp boundary. Additionally, we have shown a synthetic example here below. |
| --- | --- | --- |

(a) *Single big size sharp boundary causative synthetic model example*

(b) Inversion results of the model in (a) using the presented method.

---

## Author Comment (AC2)

| Reviewer | *Comments/corrections /Questions* | Accepted / not Accepted | Authors Response |
|---|---|---|---|
| *RC2* | **General Comments:** This work contains interesting improvements over a previous work from the authors (Gebre and Lewi, 2022) concerning gravity inversion using L0-norm regularization. The main contributions include auto- adaptive regularization and combined stopping criteria. The results shown include many tests with synthetic data and real data which supports the claim of the article. In general, the article is well written and should be accepted after few minor modifications. | Accepted | *We would like to thank reviewer* **RC2** *for the encouraging and constructive comments that contributed to the improvement of the manuscript. We highly appreciated his interesting and positive suggestions.* **Note:** *We have accommodated all of the recommendations. The applied changes are highlighted in yellow color in the revised manuscript.* |
| *RC2* | **Corrections /Comments 1:** The article contains typos and minor grammar mistakes that should be corrected. For example, in line 122, it should be "Here the misfit functional is $\Phi d =$… and $\mathbf{W}e$ is the error…". In line 334, it should be "total number of model parameters", etc… | Accepted | *We thank the reviewer for the valuable corrections. We completely agreed and incorporated the corrections.* |
| *RC2* | **Corrections Comments 2:** For the sake of completeness, the expression of the stabilizing functional, $S(\varrho)$ should be given to explicit the role of $\mathbf{W}ck$ in the objective function and improve the understandability of (4). | Accepted | *To make this point clearer as suggested by reviewer* **RC2***, we have incorporated additional* descriptions *in the revised manuscript.* |
| *RC2* | ***Corrections Comments 3:*** Also, as the method is based on iteratively reweighted least squares (IRLS), it should be mentioned on the text. | Accepted | *Yes that is true, the method is based on iteratively reweighted least squares (IRLS) minimization. As suggested, we have now mentioned it in the text.* |
| *RC2* | **Question/corrections Comments 4:** In (18), the max operation should be defined. | Accepted | *We incorporated the definition of the "max" operation.* |

| RC2 | **Corrections/Comments 5:** In (19), please use a notation similar to: $$[\tilde{\rho}^k]_j = \begin{cases} [\rho_{max}]_j \ if \ [\rho^k]_j > [\rho_{max}]_j \\ [\rho^k]_j \ if \ [\rho_{min}]_j \le [\rho^k]_j \le [\rho_{max}] \\ [\rho_{min}]_j \ if \ [\rho^k]_j < [\rho_{min}]_j \end{cases}$$ | Accepted | *As per the reviewer's suggestion, we have changed the notation in the revised manuscript.* |